# State-of-Health Estimate for the Lithium-Ion Battery Based on Constant Voltage Current Entropy and Charging Duration

Laijin Luo [1], Chaolong Zhang [1,2,*], Youhui Tian [3] and Huihan Liu [1]

1   School of Electronic Engineering and Intelligent Manufacturing, Anqing Normal University, Anqing 246011, China
2   College of Intelligent Science and Control Engineering, Jinling Institute of Technology, Nanjing 211169, China
3   School of Intelligent Engineering Technology, Jiangsu Vocational Institute of Commerce, Nanjing 211168, China
*   Correspondence: zhangcl@aqnu.edu.cn

**Abstract:** An accurate state-of-health (SOH) estimation is vital to guarantee the safety and reliability of a lithium-ion battery management system. In application, the electrical vehicles generally start charging when the battery is at a non-zero state of charge (SOC), which will influence the charging current, voltage and duration, greatly hindering many traditional health features to estimate the SOH. However, the constant voltage charging phase is not limited by the previous non-zero SOC starting charge. In order to overcome the difficulty, a method of estimating the battery SOH based on the information entropy of battery currents of the constant voltage charging phase and charging duration is proposed. Firstly, the time series of charging current data from the constant voltage phase are measured, and then the information entropy of battery currents and charging time are calculated as new indicators. The penalty coefficient and width factor of a support vector machine (SVM) improved by the sparrow search algorithm is utilized to establish the underlying mapping relationships between the current entropy, charging duration and battery SOH. Additionally, the results indicate the adaptability and effectiveness of the proposed approach for a battery pack and cell SOH estimation.

**Keywords:** lithium-ion battery; state of health; state of charge; current entropy; sparrow search algorithm; support vector machine

## 1. Introduction

Lithium-ion batteries play an increasingly important role in many fields, such as energy storage, aviation, aerospace and new energy vehicles, owing to the battery's prominent advantages of a long-cycle life, low self-discharging rate, low cost and low environmental pollution [1,2]. The state of health (SOH), as one of the key indicators, is usually to measure the degree of the battery aging and whether the battery requires to be changed in time [3]. However, the repeated charge–discharge cycle of a battery will cause many serious problems, such as fast power consumption, long charging time, shortened driving distance, etc. Therefore, it is necessary to accurately estimate the battery SOH.

In order to obtain the aging condition of the battery more quickly and accurately, many battery SOH estimation methods were proposed in the past few years. These methodologies can be divided into three categories: (1) the experimental estimation approaches [4,5], (2) the model-based methods [6–10] and (3) the data-driven methods [11–15]. Experimental estimation approaches directly acquire specific characteristic parameters by measuring equipment and further analyze the degradation of the battery performance. Model-based methods simulate the internal structure, materials and chemical reactions of a battery by establishing an equivalent model. Data-driven methods only need to analyze historical data from laboratory measurements [16] and do not have to analyze the mechanism of the internal electrochemical reaction of the battery. It can directly establish the mapping

relationship between the current, voltage, other characteristics and maximum discharge capacity, and then estimate the battery SOH. For the experimental estimation and model-based methods, a stable experimental environment or complex experimental equipment is required, which hinders their application. In recent years, the data-driven methods of the SOH estimation have attracted more and more scholars' attention because of their flexibility and versatility.

Multiple SOH characteristics are often used as input for data-driven approaches. For instance, Zhang et al. [17] adopted the Coulomb counting and differential voltage in a constant current to extract characteristic. Although the computational burden of the Coulomb counting formula is small, the current measurement is not only affected by the equipment precision but also by the measurement noise. From the perspective of the chemical aging mechanism, Fu et al. [18] adopted impedance characteristics as an extreme learning machine (ELM) input, which can efficiently estimate the SOH, but the result of the estimation has a lot of randomness because the input weight of the extreme learning machine is not updated by feedback. In addition to the above perspectives of multiple SOH characteristics for estimating the SOH, Li et al. [19] established a novel capacity and the remaining useful life prediction model by employing the Elman and long short-term memory neural networks, which realized a stronger robustness and generalization. Similarly, Lin et al. [20] applied a novel estimation model established by fusing multiple neural networks, and the hybrid model has strong robustness. However, the computational burden is aggravated, with the hyperparameters of the fusion model becoming more complex.

In short, some inevitable fatal drawbacks still exist in the data-driven approaches. Firstly, some methods have complex parameters as well as a poor prediction ability for small sample data. Secondly, the neural network parameters are easily caught in the local optimum, which will create the overfitting problem [21]. The support vector machine (SVM) is one of the frequently employed data analysis algorithms. It can map nonlinear data to linear data by kernel function to proceed nonlinear system modeling. However, the parameters need to be optimized to maximize the estimation performance; the sparrow search algorithm (SSA) is derived from the behavior of sparrows foraging and escaping predators, and it has a comparatively good local search ability [22].

Additionally, the voltage data or current data in the whole phase of the constant current–constant voltage (CC-CV) charging, starting from the zero state of charge (SOC), is generally used in many studies on the SOH estimation of batteries to carry out the state estimation [23,24]. However, with new energy vehicles, in the actual application, charging starts when the battery SOC drops to 30–80%, so it is hard to extract the charging data with a starting point at zero SOC for the SOH estimation of the battery. The CV charging phase is the last stage; the maximal charging voltage of the battery cell or battery pack is maintained, and the current is slowly decreased to cut off the current in this phase. Therefore, the current sequence and charging time are extracted from the constant voltage–current curve as new features. Information entropy can quantify the current sequence into a value, which can effectively avoid the measurement error and cut down the computational burden.

To overcome the above-mentioned problems, a method for estimating the battery SOH, based on the information entropy of battery currents of the CV charging phase and charging duration, is proposed. Firstly, the charging currents data of a battery cell or pack from the constant voltage charging phase are extracted, and then the information entropy of the battery currents and charging time are calculated as the input features. The penalty coefficient and width factor of the SVM optimized by the sparrow search algorithm is utilized to establish the mapping relation between the battery SOH, charge duration and current entropy. After building up the SOH estimation model of the battery, it can quickly and efficiently estimate the battery pack and cell SOH value. Specifically, several key contributions are presented below.

(1) An original SOH combination indicator is proposed to estimate the battery SOH when it starts charging at a non-zero SOC. By employing information entropy to quantify

the current sequence, the current entropy and charging duration is deduced from the CV–charging current curve.

(2) The computational burden and precision of the SOH estimation are compared with four other traditional methods of employing a different number of input features. Although the amount of the calculation burden of the proposed method increases, the precision of the SOH estimation value has been greatly improved.

(3) The adaptability and effectiveness of the proposed approach for the battery pack and cell SOH estimation are verified based on two different types of batteries: battery cell and battery pack.

## 2. The Battery SOH

The battery is tested under a laboratory environment with repeated charge–discharge cycles, which will lead to the battery's maximum available capacity loss. Therefore, the SOH of lithium batteries is defined by capacity in this paper.

$$\text{SOH} = \frac{Q_{\text{max capacity}}}{Q_{\text{rated capacity}}} \times 100\% \tag{1}$$

where $Q_{\text{max capacity}}$ is the maximum discharge capacity at present, and $Q_{\text{rated capacity}}$ is the factory-specified rated capacity of the battery.

## 3. Feature Extraction

In practice, the CC-CV charging is a common charging mode, which not only has high charging efficiency but also causes less damage to the battery. A larger current can be used in the CC charging phase, so that the charging efficiency will be relatively high. However, the polarization internal resistance will lead to a high virtual voltage of the battery. Therefore, it is necessary to add the CV charging phase, which can make the battery's SOC fuller. In this paper, each charging process includes two phases, which are showed in Figure 1. Specifically, in the CC phase, the charging current is constant until the voltage rises to the upper cut-off voltage, and then in the CV phase, the voltage is kept constant until the current slowly reduces to the lower cut-off current. The current sequence in the CV phase is selected as the feature.

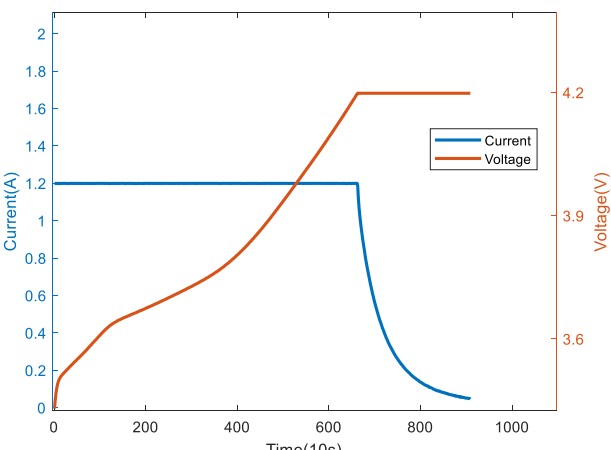

**Figure 1.** Constant current–constant voltage charging curve.

## 3.1. Current Entropy

Entropy originates from physics and is usually used to measure the stability of the thermodynamic systems. The lower the entropy is, the more ordered the system is. On the contrary, the higher the entropy is, the more disordered the system is. In discrete data, information entropy is the probability of discrete random events, which can be considered

as a measure of information. Similarly, the discrete data are more ordered, the information entropy is smaller. In the opposite way, the discrete data are more disordered, the information entropy is larger. The information entropy formula [25] is defined as following.

$$H(x) = -\sum_{x=1}^{n} p_x \log_2(p_x) \tag{2}$$

where $p_x$ represents the probability of appearance of sampling current of value $x$ for all $n$, and $n$ is the number of sampling currents.

### 3.2. Charging Time

The battery is tested under a complex and variable environment with repeated charge–discharge cycles, and internal irreversible chemical reactions, which will cause the increase in resistance with increasing cycle times, and its direct characterization of aging can be seen in the charging curve, as shown in Figure 2. The initial CC charging duration of the newly produced battery is the longest. However, with increasing cycle times, the CC charging duration is constantly decreasing, and the CV charging duration is gradually increasing. It means that the charging time is associated with the battery SOH [26,27]. Therefore, either the CC charging time or CC charging time can be selected as the input feature.

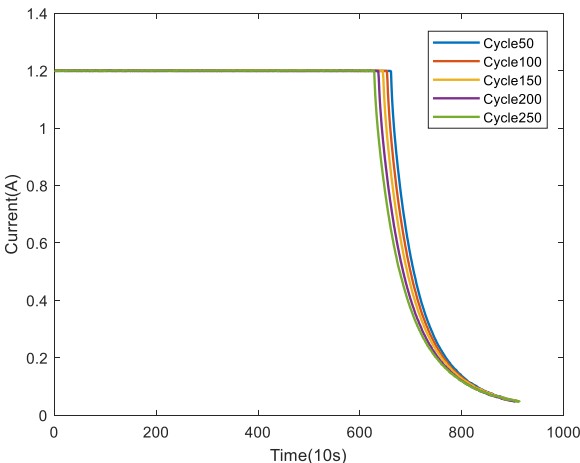

**Figure 2.** Charging–time curves at different cycles.

## 4. SSA-SVM

### 4.1. Support Vector Machine

SVM reorganizes nonlinear and disordered data into linear space for classification. It not only keeps fast and accurate estimation results even under conditions of the nonlinear or small sample data but also achieves fast convergence effect by reducing the complex calculation [28]. The function of that transformation process is expressed as follows:

$$f(x, w) = w \cdot \varphi(x) + b \tag{3}$$

where $w$ is the vector of weight of input quantity, $b$ is the bias of regression and $\varphi$ is nonlinear mapping function. Optimal planning expressions:

$$\begin{cases} \min f(w) = \min\frac{1}{2}\| w \|^2 + c\sum_{i=1}^{n} e_i^2 \\ s.t \ \ w\varphi(x_i) + b \geq y_i - e_i, i = 1, 2, \cdots, n \end{cases} \tag{4}$$

where $c$ is a penalty coefficient of the error term, $e_i$ represents the error value of regression. The Lagrange equation is established through the above constraint conditions:

$$L(w, b, e, \mu) = \frac{1}{2}\| w \|^2 + c\sum_{i=1}^{n} e_i^2 - \sum_{i=1}^{n} \mu_i(w\varphi(x_i) + b + e_i - y_i) \tag{5}$$

where $u_i$ represents Lagrange multiplier. After finding first-order partial derivative of each variable based on Karush–Kuhn–Tucker (KKT) equations, arranging and converting the optimal solution into the following linear equations.

$$\begin{bmatrix} 0 & I_n^T \\ I_n & K + \frac{I_n}{c} \end{bmatrix} \begin{bmatrix} b \\ u \end{bmatrix} = \begin{bmatrix} 0 \\ y_n \end{bmatrix} \tag{6}$$

where $I_n$ is a unit vector of order $n$, $u = [u_1, u_2, \dots, u_n]$, $K$ is the matrix of kernel function and it is described as follows:

$$K(x_i, x_j) = \exp(-\frac{\| x_i - x_j \|^2}{2\sigma^2}) \tag{7}$$

where $\sigma$ is a width factor for controlling the shape of the kernel function. The mapping function of SVM is described as follows:

$$f(x) = \sum_{i=1}^{n} (\mu_i K(x, x_i)) + b \tag{8}$$

In SVM algorithm, the penalty factor and width factor are two critical parameters, which need to be set as optimal values to reach the best estimation effect. Therefore, the SSA is used to optimize above two parameters in this paper.

### 4.2. Sparrow Search Algorithm

According to their ability of foraging, the sparrow population of SSA is divided into producers, scroungers, who have an awareness of danger. In addition, between 10 and 20 percent of these sparrows are presumed to have a sense of danger, which are selected as defenders [22]. The SSA iteratively updates the location information of the producers, scroungers and defenders based on the individual fitness, and then obtains the location information of the global optimal. The location information of each sparrow is updated in the multidimensional space, and the update process is described as follows.

Producer's location update state:

$$X_{i,j}^{t+1} = \begin{cases} X_{i,j}^t \cdot \exp(\frac{-i}{\alpha \cdot T_{\max}}), & R_2 < ST \\ X_{i,j}^t + Q \cdot L, & R_2 \geq ST \end{cases} \tag{9}$$

where $X_{i,j}^{t+1}$ is the $j$th-dimension position information of the $i$th sparrow in $t + 1$th generation, $T_{\max}$ is the maximum iterations, $\alpha$ and $R_2 \in (0, 1]$, $ST \in [0.5, 1.0]$, $Q \in [-1, 1]$ is a random number, $L$ denotes a matrix.

Scrounger's location update state:

$$X_{i,j}^{t+1} = \begin{cases} Q \cdot \exp(\frac{X_{worst}^t - X_{i,j}^t}{i^2}), & i > \frac{n}{2} \\ X_p^{t+1} + \left| X_{i,j}^t - X_p^{t+1} \right| \cdot A^+ \cdot L, & i \leq \frac{n}{2} \end{cases} \tag{10}$$

where $X_p^{t+1}$ is the best location information in $t + 1$th generation, $X_{wrost}^t$ represents the worst location information in $t$th generation, $A^+$ denotes a matrix.

Defender's location update state:

$$
X_{i,j}^{t+1} = \begin{cases} X_{best}^{t} + \beta \cdot \left| X_{i,j}^{t} - X_{best}^{t} \right|, f_i > f_g \\ X_{i,j}^{t} + K \cdot \left( \dfrac{\left| X_{i,j}^{t} - X_{worst}^{t} \right|}{(f_i - f_w) + \varepsilon} \right), f_i = f_g \end{cases}
\tag{11}
$$

where $X_{best}^{t}$ is the best location information in $t$th generation, $\beta$ and $K \in [-1, 1]$ are both random numbers, $f_i$ represents the fitness of the $i$th sparrow, $f_g$ and $f_w$ represent the best and worst fitness of the current generation, respectively, and $\varepsilon$ denotes the small number.

### 4.3. The Process of SSA Optimizing SVM

The width factor and penalty coefficient of SVM is improved by SSA, and mean square error is defined as individual fitness. Compared with the traditional algorithm, the SSA not only has a better ability to search the best location information but can also effectively avoid local optimal issue. Such as the ant colony optimization algorithm search ability is slowly, the convergence value of particle swarm optimization algorithm will be obtained too early. The flowchart of the SSA-SVM is shown in Figure 3. Furthermore, the penalty coefficient and width factor are mapped to the two-dimensional location information of each sparrow. After optimizing two-dimensional location information, the penalty coefficient and width factor of the optimal two-dimensional location information are applied to the SVM. The detailed processes of SSA to single out the best values for penalty coefficient and width factor of SVM are described as follows.

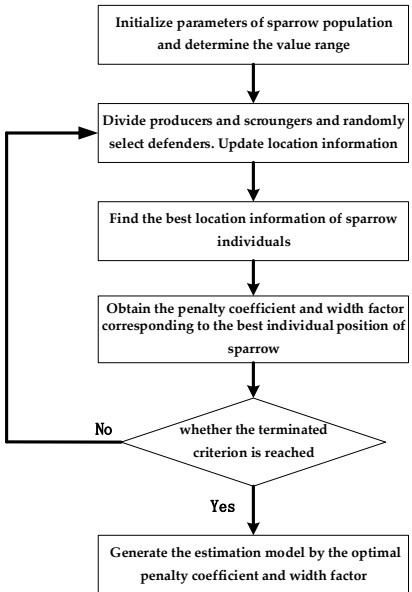

**Figure 3.** The flowchart of the SSA-SVM.

Step 1: Initialize parameters of sparrow position information. Meanwhile, determine the value range of penalty coefficient and width factor.

Step 2: Divide producers and scroungers and randomly select defenders. Update lo-cation information of producers, scroungers, defenders, respectively.

Step 3: Find the best location information of sparrow individuals based on the principle of SSA.

Step 4: Obtain the penalty coefficient and width factor corresponding to the best in-dividual position of sparrow.

Step 5: Repeat step (2)~(4) to judge whether the terminated criterion is reached, that is, the maximum number of iterations.

Step 6: The SSA-SVM estimation model is established by the optimal penalty coefficient and width factor are assigned to the SVM for training.

## 5. Experimental Steps and Results Analysis

### 5.1. Battery Data

The battery is tested under a laboratory environment with repeated charge–discharge cycles. Specifically, in the CC phase, the charging current is maintained until the voltage rises to the upper cut-off voltage, and then the voltage is kept until the current is reduced to the lower cut-off current in CV phase. Finally, the discharging current is constant until the voltage drops to the lower cut-off voltage. The experimental data of battery pack and cell are obtained in AQNU university laboratory, and the SOH measurements of battery pack and cell are showed in Figure 4. The battery pack consists of six same 18,650 lithium-ion cells linked in series. The aging data of battery pack and cell are measured at room temperature with repeated charge–discharge cycles, by the efficient battery test system assembled from laboratory equipment, which includes the upper machine, battery pack measuring equipment, cell measuring equipment, incubator. The upper machine is used to record and store data of battery. Battery pack and cell measuring equipment are provided with the constant current, constant voltage and other settable charging and discharging modes. The efficient battery test system is shown in Figure 5.

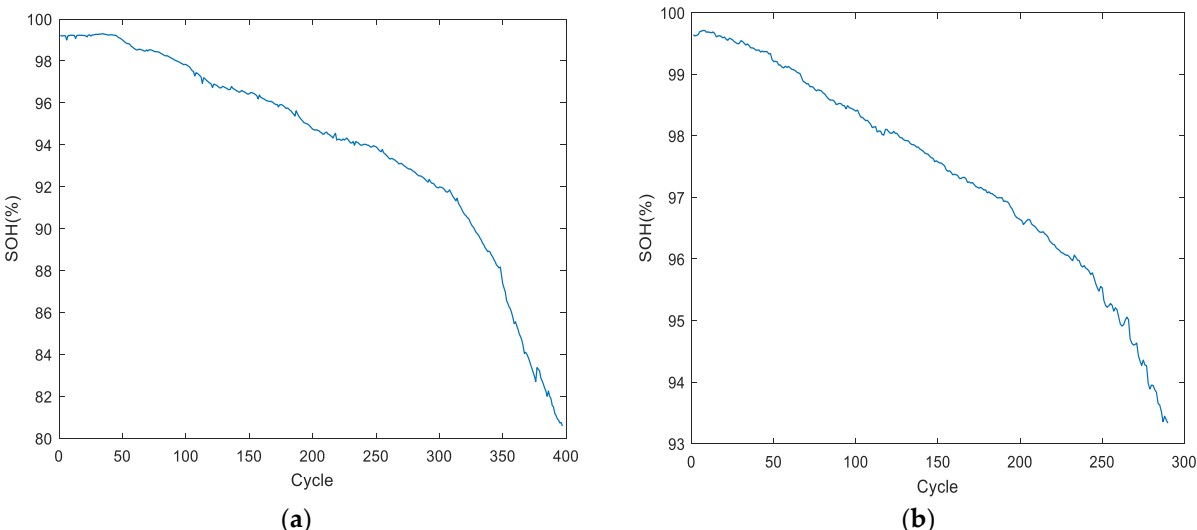

**(a)**                    **(b)**

**Figure 4.** Measured SOH data of battery pack and cell. (**a**) Battery pack; (**b**) battery cell.

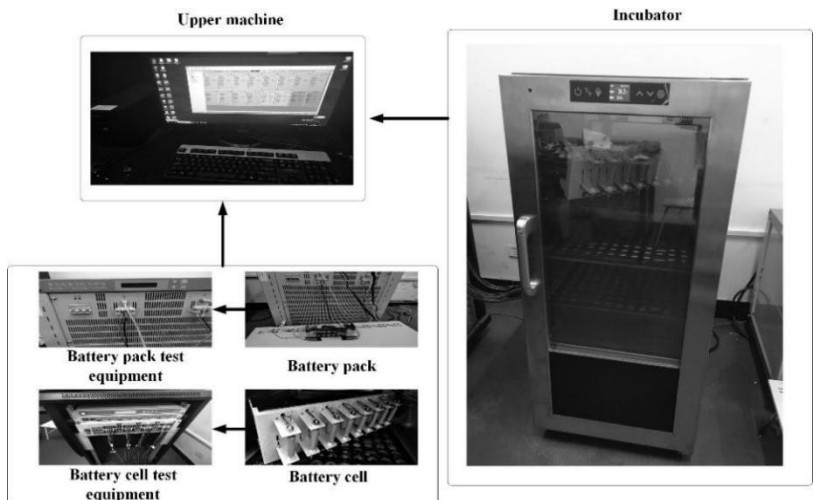

**Figure 5.** The efficient battery test system.

## 5.2. Experimental Procedures

In this experiment, the SOH of battery pack and cell is estimated, respectively. The detailed steps are presented below.

Step 1: Extract charging currents data of battery pack and cell from each constant voltage charging phase, and then calculate the information entropy of battery currents and charging time from each charging cycle; the SOH data are from each discharging cycle.

Step 2: The calculated indictors are divided into training set and test set according to 1:1.

Step 3: Based on the training set, using SSA to optimize the width factor and penalty coefficient of SVM.

Step 4: The optimal penalty coefficient and width factor are employed to establish SOH estimation models of the battery pack and cell, respectively.

Step 5: Estimate SOH of battery pack and cell based on the test set.

## 5.3. Experimental Results Analysis

When the SSA optimizes the parameters of the SVM, the mean square error (MSE) between the estimated SOH and the actual SOH is adopted as the fitness. Taking the best fitness value from the sparrow populations of each iteration until the end of the iteration, the number of iterations of the SSA is 100. The penalty coefficient and width factor of the SVM are optimized by the SSA. The iterative optimization processes are shown in Figure 6.

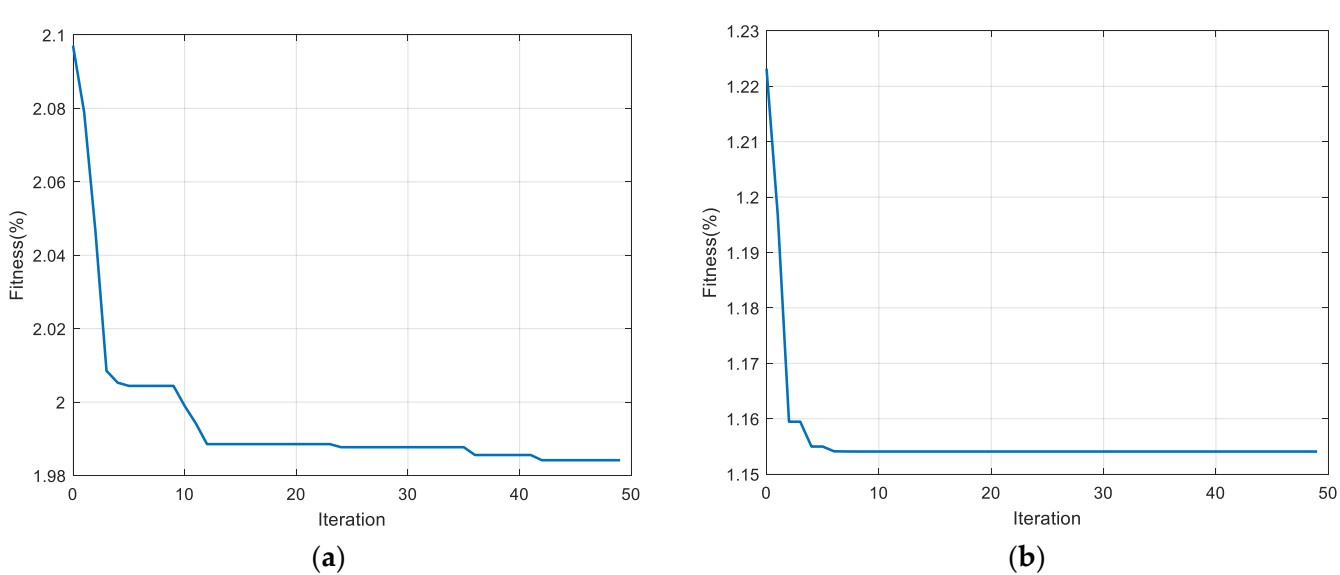

**Figure 6.** Iterative optimization processes of battery pack and cell. (**a**) Battery pack; (**b**) battery cell.

The penalty coefficient and width factor of the SVM optimized by the SSA are shown in Table 1. The optimized penalty coefficient and width factor by the SSA are applied, and then the battery pack and cell SOH is estimated by employing the SVM, respectively. The results are shown in Figure 7, by the proposed method of the battery pack and cell SOH estimation which are very close to the SOH measurement, indicating that the proposed method based on the information entropy of the battery currents of the constant voltage charging phase and constant voltage charging duration as input features is valid. In addition, the bar error chart of the battery and cell are shown in Figure 8, indicating the proposed method has a high accuracy. From the bar error chart, it can be seen that the bar error of the battery pack SOH estimation is within 1.3%. Furthermore, the bar error of the battery cell SOH estimation is within 0.7%. Therefore, the proposed method is effective, which can exactly estimate the battery pack and cell SOH.

**Table 1.** The optimized penalty coefficient and width factor of SVM in the proposed method.

| Battery Type | Penalty Coefficient | Width Factor |
|---|---|---|
| Battery pack | 24.5285 | 0.0914 |
| Battery cell | 6.7407 | 0.0610 |

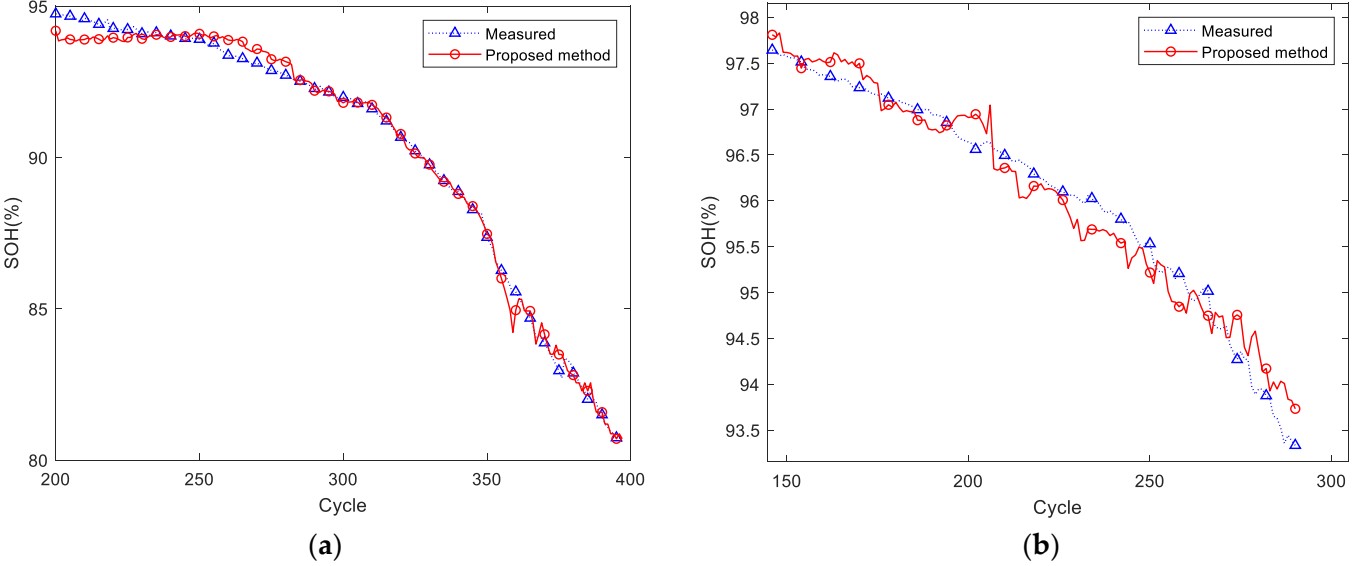

**Figure 7.** The estimation results of SSA-SVM. (**a**) Battery pack; (**b**) battery cell.

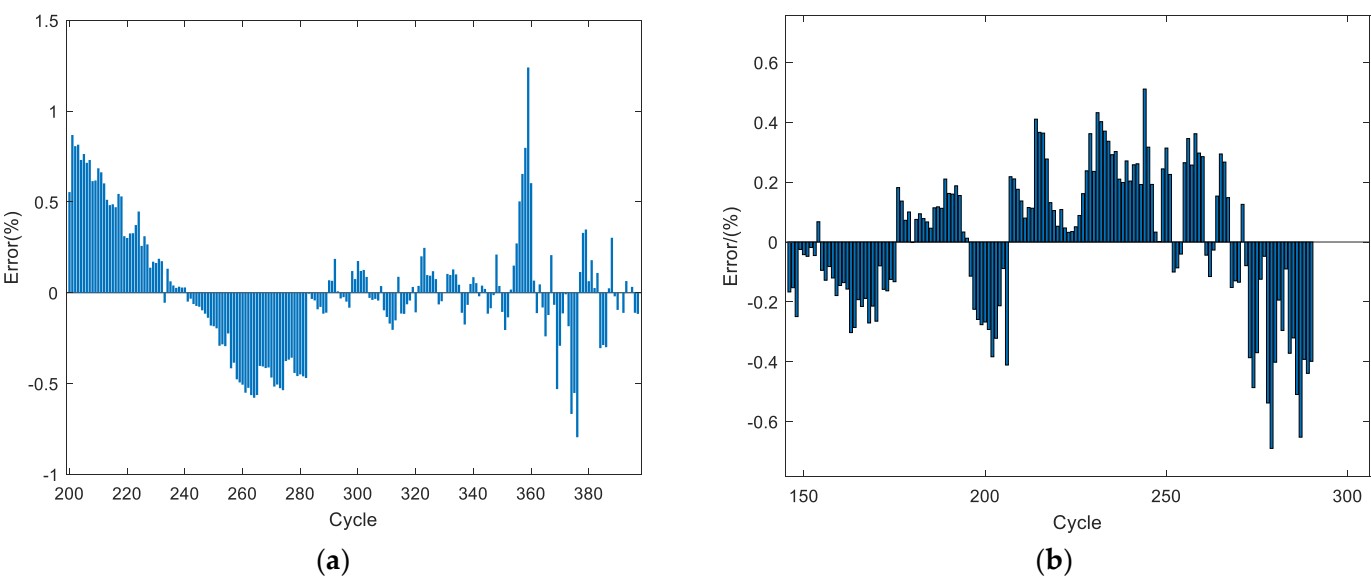

**Figure 8.** The SOH bar errors between estimation and measurement. (**a**) Battery pack; (**b**) battery cell.

In order to prove the feasibility of the SOH estimation method, we set up four comparative experiments. Comparative experiment 1 was observed from the perspective of the dimension of the input features. Comparative experiment 2 contrasted from the perspective of whether to use the algorithm to optimize the parameters. Finally, comparative experiments 3 and 4 were both from the perspectives of a different network. The specifics of the comparative experiment are displayed in Table 2, and the comparison results of the comparative experiment are shown in Figure 9.

**Table 2.** The specifics of the comparative experiment.

| Method | Input | Estimation Method |
|---|---|---|
| Proposed method | Current entropy and charging time | SSA-SVM |
| Compared method 1 | Current entropy | SSA-SVM |
| Compared method 2 | Current entropy and charging time | SVM |
| Compared method 3 | Current entropy and charging time | Elman |
| Compared method 4 | Current entropy and charging time | ELM |

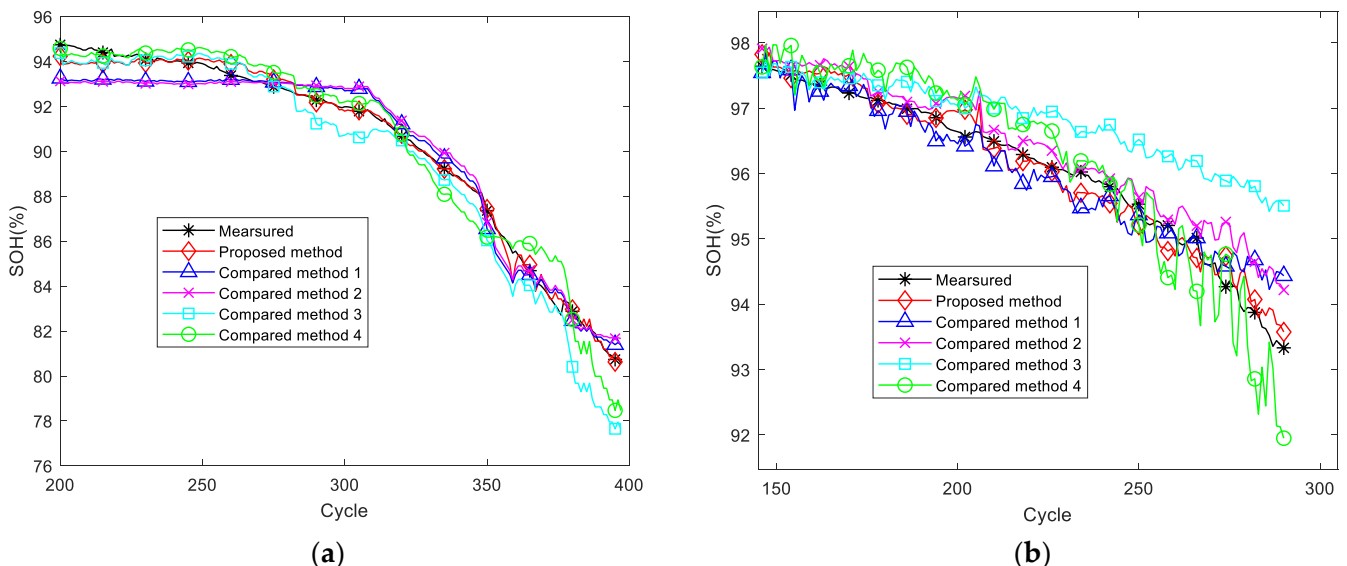

**Figure 9.** The comparison results of comparative experiment. (**a**) Battery pack; (**b**) battery cell.

The absolute error between the battery SOH estimation values and the measurement SOH values are shown in Figure 10. The mean absolute error (MAE), maximal error (ME) and computational burden of the comparative experiment are displayed in Table 3. The proposed method is compared with method 1 from the perspective of the dimension of input features, the proposed method provides a better prediction accuracy and only slightly increases the calculation cost. The proposed method is compared with method 2 from the perspective of parameter optimization. After optimizing the penalty coefficient and width factor of the SVM by the SSA, the calculation cost of the proposed method increases, but the estimation precision is greatly improved. The SSA is good at searching the global optimization, and the local optimal issue can be avoided effectively. Regarding the comparison by method 3 and method 4, the Elman neural network has the advantages of local memory and feedback and can better estimate the battery SOH; however, the input weight of the ELM is random, and there is no need to learn and update the input weight. It can be observed that the Elman neural network has much more computational cost than the ELM, but the accuracy of the SOH estimation is lower than the ELM. Therefore, among the SOH estimation methods for the battery pack and cell, the SSA-SVM has a higher accuracy, and the ELM is more suitable for estimating the SOH than the Elman neural network in this paper.

In compared method 1, the SSA-SVM adopts the current entropy of each constant voltage charging phase as the feature input in comparative experiment 1, which the MAE and ME are 0.65 and 1.57%, respectively. However, the MAE and ME of the proposed method are 0.24 and 1.2%, respectively, which are all lower than the MAE and ME of compared method 1. Because the charging duration of the constant voltage charging phase is not considered in comparison method 1, and the battery is tested under a complex and variable environment with repeated charge–discharge cycles, there are resulting internal irreversible chemical reactions. Therefore, it can be considered that the increase in the

charging duration is associated with the aging process, and the experimental results prove that the charging duration is an important feature of battery aging. Similarly, it is also shown that the constant voltage charging duration is related to the SOH in Ref. [29].

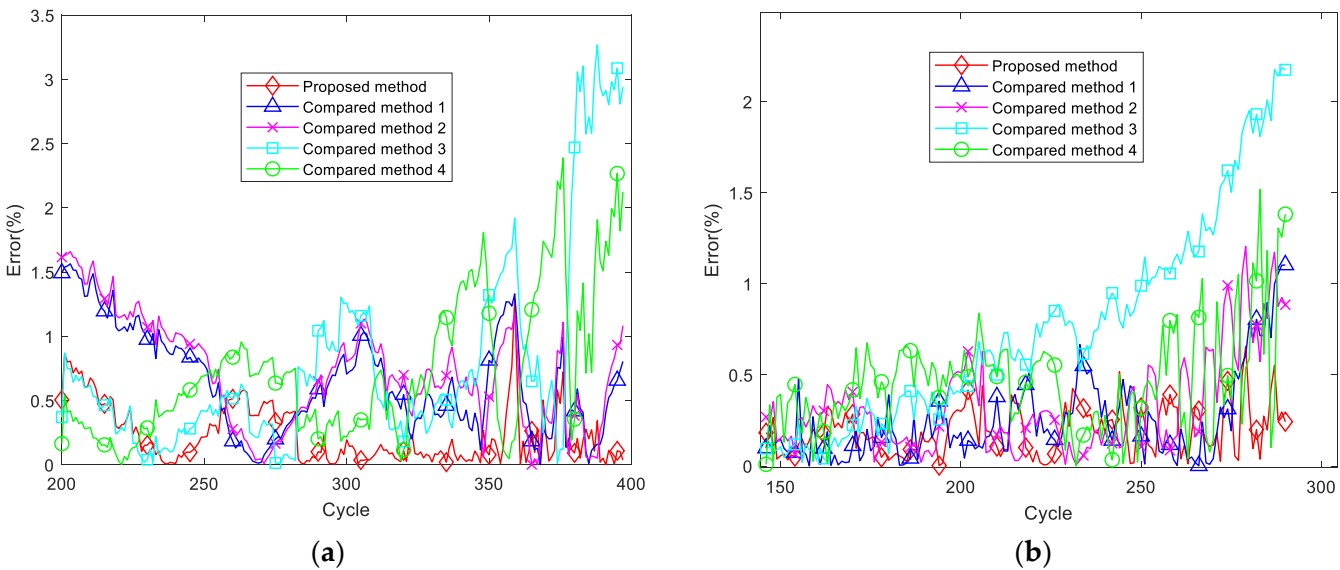

**Figure 10.** The absolute SOH estimation error values of the comparative experiment. (**a**) Battery pack; (**b**) battery cell.

**Table 3.** The MAE, ME and computational burden of the comparative experiment.

| Method | Error | Battery Pack | Battery Cell | Time (s) |
|---|---|---|---|---|
| Proposed method | MAE (%) | 0.2364 | 0.1939 | 7.3965 |
| | ME (%) | 1.2924 | 0.6502 | |
| Compared method 1 | MAE (%) | 0.6482 | 0.2525 | 7.3620 |
| | ME (%) | 1.5650 | 1.1040 | |
| Compared method 2 | MAE (%) | 0.7139 | 0.3251 | 0.0121 |
| | ME (%) | 1.6599 | 1.2058 | |
| Compared method 3 | MAE (%) | 0.7963 | 0.7426 | 4.0970 |
| | ME (%) | 3.2696 | 2.1865 | |
| Compared method 4 | MAE (%) | 0.7212 | 0.4594 | 0.0105 |
| | ME (%) | 2.3906 | 1.5180 | |

In compared method 2, the SVM adopts the current entropy and charging duration of each constant voltage charging phase as the feature input, but the MAE and ME are larger. Owing to the penalty factor and width factor playing an important role in the estimation performance of the model, the penalty coefficient is used to control the weight of loss and the width factor influences the radial action range of the kernel function and determines the distribution characteristics of the training sample data. Therefore, in this paper, the SSA is used to optimize the penalty coefficient and width factor of the SVM, which can effectively improve the precision of estimating the SOH.

In compared method 3, the Elman neural network of the local memory and feedback is used to estimate the battery SOH, MAE and ME of the Elman neural network which are 0.8 and 3.27%, respectively, which are all higher than the MAE and ME of the proposed method. This is because an SVM can map nonlinear problems to linear problems by the kernel function [30]. In the case of a small sample size, the global optimization can still be searched, without the weaknesses of training slowly and falling into local optimization

caused by the gradient method. Therefore, the SSA-SVM method based on current entropy and charging duration has better accuracy in estimating the battery SOH in this paper.

In compared method 4, the extreme learning machine is used and compared with the classical neural network. The input weight of the extreme learning machine is random, and there is no need to update the input weight of the hidden layer [31]. However, the Elman neural network has feedback; it needs to learn and update the weight by the gradient method [32]. Therefore, the ELM has a fast calculation output and strong generalization ability. Its MAE and ME are 2.39 and 0.72%, respectively, which can quickly and accurately estimate the battery SOH, but it is inferior to the ability of the SVM to process nonlinear data.

In conclusion, the precision of the SSA-SVM estimation SOH method based on the current entropy and charging duration is higher than the other estimation methods. In comparative experiment 1, the MAE and ME are both lower, which can indicate that composite features of current entropy and charging duration are superior to the single feature in the SOH estimation. Similarly, the MAE and ME of comparative experiments 2, 3 and 4 are all higher than the proposed method, which also indicate that the accuracy of the SSA-SVM is higher than other estimation methods.

## 6. Conclusions

A novel method for estimating the SOH has been proposed based on the information entropy of the battery currents of the constant voltage charging phase and constant voltage charging duration. After obtaining the battery currents and charging time from each constant voltage charging phase, the current entropy and charging time are calculated as features to characterize the aging process. The penalty coefficient and width factor of the SVM are optimized by the SSA, and then the optimal mapping relationship between the battery SOH, charge duration and current entropy is obtained, and the battery SOH estimation model is constructed. Furthermore, in order to prove the validity of the proposed method, the error between the aging data SOH measured by the laboratory equipment and the estimated SOH has been analyzed, and then four groups of comparative experiments have been set up. The comparative experiment results indicate the adaptability and effectiveness of the proposed approach, and the maximal error is within 1.3%. In addition, the proposed method is not only suitable for the SOH estimation of the battery pack but also for the battery cell. Therefore, it has strong guidance and reference in SOH estimation methods of the battery pack and cell.

**Author Contributions:** Validation, formal analysis and writing, L.L.; methodology and software, C.Z.; conceptualization, Y.T.; project administration, H.L. All authors have read and agreed to the published version of the manuscript.

**Funding:** This work was supported by the "Four New" research and reform practice project under Grant No. 2021sx092.

**Institutional Review Board Statement:** Not applicable.

**Informed Consent Statement:** Not applicable.

**Data Availability Statement:** The datasets that support the findings of this study are measured in the corresponding author's laboratory and available from the corresponding author upon reasonable request.

**Conflicts of Interest:** The authors declare no conflict of interest.

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
