# Peer review of "State-of-Health Estimate for the Lithium-Ion Battery Based on Constant Voltage Current Entropy and Charging Duration"

_wevj, doi:10.3390/wevj13080148_

Round 1

Reviewer 1 Report

The paper addresses a topic of great interest. It's well written and there is some kind of experimental validation that helps. I have a few comments though:

1. The references can be significantly improved by including more and also recent papers that focus on he same topic. An example can be:  Misyris, George S., et al. "Battery energy storage systems modeling for online applications." 2017 IEEE Manchester PowerTech. IEEE, 2017.

2. The paper needs to investigate more aspects and the content needs to become bigger. More tests, more benchmarking against published literature, investigation of other battery types can be ideal.

3. Which are the papers behind the compared methods 1-4 mentioned in the manuscript?

4. Please describe in more detail the experimental & lab setup

5. What can you discuss about computational effort/burden?

Author Response

Thank you very much for your comments and suggestions. You have pointed out our mistakes in the paper and your comments are useful and valuable. Through revision, the paper improves its quality and readability. The responses are in the WOR file.

Reviewer 2 Report

The review report of manuscript wevj-1834651, Titled, “State-of-health estimate for the lithium-ion battery based on constant voltage current entropy and charging duration”

This paper introduces an experimental work to estimate the state of health of the lithium ion batteries using a support vector machine (SVM) optimized by sparrow search algorithm (SSA).

This paper is not introducing any novel work that can add any value to the readers and it needs substaintial improvement before it can be considered for publication as shown in the following points:

The authors should highlight the main contribution and novelity introduced in the abstract section and how this improvement to the battery SOH estimation is outperforming the previous efforts in this issue.

The introduction part should define the problem and its importance to the scientific community. Moreover, the author should discuss the previous technique used in this regard and discuss the limitations associated with these techniques and how they overcome these limitations in their methodology.

The authors should highlight the mathematical and imperical manupolation of the equation used to predict the SOH of the battery. And they should discuss the need for using constant current and constant voltage operating conditions for SOH determination.

The authors should mention how they measue the experimental results and how they controlled the batteries to work in CC and CV modes.

The flowchart showing the SVM in the SSA optimization algorithm should be included in the results and they should mention the scientific reason of using SSA instead of other optimization algorithms.

The mathematical equations of the SVM and SSA should be included in the paper.

Author Response

Thanks to you for your kind comments. Your comments point out the shortcomings of our paper. Meanwhile, your comments are helpful and valuable. Through revision, the paper improves its quality and readability. The responses are in the WORD file.

Reviewer 3 Report

Reviewer’s report:

Title: State-of-health estimate for the lithium-ion battery based on constant voltage current entropy and charging duration.

In this work, the authors reported a battery state-of-health estimation based on information entropy of battery current of constant voltage charging phase and constant voltage charging duration. Generally, the manuscript has too many errors without enough analysis and discussion. The manuscript presentations are not accurate in many parts.

1. Typos: “Coltage” (at Fig.1 of Y axis); value 400 at Fig.3(a) of x axis is missing; mothed → in many places of Fig.5, table 1, Fig. 7(a)-(b), Fig. 8(a)-(b), table 2, at lines 186,191,199,207.; at line 165 “compere”;

2. At line 100 to 101, the author mentioned that with increasing of cycle times, the constant current charging duration is constantly decreasing and the constant voltage charging duration is gradually increasing, why?

3. At lines 154 to 155, these sentences are not clear. “respectively” refers to what?, those sentences should be modified.

4. At lines 154 to 156, a single paragraph only consist of 3 lines is too less, add more!

5. At line 158, you mentioned “It can be seen intuitively…”, data analyzing should be in a rationally manner instead of intuitive!

6. At lines 158 to160, the author claimed that the proposed method of SOH estimation of battery based on information entropy of battery current of constant voltage charging phase and constant voltage charging duration is valid. What the reasons and indications to support your claim?

7. At Fig. 5(a)-(b), why the graphs between the proposed method and the measurement are not consistent?

8. Discussion and analyses for Figs. 5 and 6 are too less.

9. Figs. 7 and 8 are still without analysis. (What means of those graphs?, Why SOH graphs and errors graphs are not consistent?, etc).

10. At line 173, you mentioned “It can be seen from table three …”, in fact you don’t provide table.3!. There is no caption to show the table 3.

11. References are too less which only is used in the introduction. Your discussion should be in deep analysis objectively.

12. In general, the data result discussions are still shallow without any references support.

Author Response

Many thanks to you for your comments again. Your comments are very valuable and effective for our manuscript. Through revision, the quality and readability of manuscript is improved. The responses are in the WORD file.

Round 2

Reviewer 1 Report

no further comments

Author Response

Thank you for for the guides with our manuscript.  We have carefully checked some grammar errors in our manuscript and have made correction which we hope meet with approval. Revised portion are marked in red in the paper. The main corrections in the paper are as following

 1)

  unmodified: To overcome the above-mentioned problem, a battery SOH estimation method is proposed based on information entropy of battery currents of constant voltage charging phase and constant voltage charging duration.

  modified: To overcome the above-mentioned problems, a battery SOH estimation method is proposed based on information entropy of battery currents of constant voltage charging phase and constant voltage charging duration.

  2)

  unmodified: Applying the optimized penalty coefficient and width factor by SSA, SVM is used to estimate battery pack and cell SOH respectively.

  modified: The optimized penalty coefficient and width factor by SSA are applied, and then SVM is used to estimate battery pack and cell SOH respectively.

  3)

  unmodified: In this paper, a battery SOH estimation method is proposed, based on in-formation entropy of battery currents of constant voltage charging phase and constant voltage charging duration.

  modified: In this paper, a battery SOH estimation method has been proposed based on in-formation entropy of battery currents of constant voltage charging phase and constant voltage charging duration.

  4)

  unmodified: In order to prove the validity of the proposed method, the error analysis is made be-tween the aging data SOH measured by laboratory equipment and the estimated SOH, and four groups of comparative experiments are set up.

  modified: In order to prove the validity of the proposed method, has been analyzed error between the aging data SOH measured by laboratory equipment and the estimated SOH, and then four groups of comparative experiments have been set up.

  5)

  unmodified: Proposed method is not only suitable for SOH estimation of the battery pack, but also for battery cell.

  modified: Besides, the proposed method is not only suitable for SOH estimation of the battery pack, but also for battery cell.

Reviewer 2 Report

The authors covered all my previous comments and I do not have any other comments except a final revision for English grammar and typos.

Author Response

(The authors gave the same response as above.)

Reviewer 3 Report

Reviewer’s report:

Title: State-of-health estimate for the lithium-ion battery based on constant voltage current entropy and charging duration.

The manuscript has been revised properly by authors. Discussion and analysis have been added comprehensively. The manuscript presentation has been improved significantly. Overall, it may will beneficially for other scientists who works in the field. However, it still needs revision at some small errors below.

1. At lines 116 to 117, typo: … two different types of batteries: battery pack and battery pack.

2. At line 196, typo:  

Author Response

(The authors gave the same response as above.)
